# Effect of Periodontal Disease on Diabetic Retinopathy in Type 2 Diabetic Patients: A Cross-Sectional Pilot Study

**DOI:** 10.3390/jcm9103234

**Published:** 2020-10-09

**Authors:** Yuko Yamamoto, Toshiya Morozumi, Takahisa Hirata, Toru Takahashi, Shinya Fuchida, Masami Toyoda, Shigeru Nakajima, Masato Minabe

**Affiliations:** 1Department of Dental Hygiene, Kanagawa Dental University, Junior College, 82 Inaoka, Yokosuka 2388580, Kanagawa, Japan; yamamoto.yuko@kdu.ac.jp; 2Division of Periodontology, Department of Oral Interdisciplinary Medicine, Graduate School of Dentistry, Kanagawa Dental University, 82 Inaoka, Yokosuka 2388580, Kanagawa, Japan; t.hirata@kdu.ac.jp (T.H.); minabe@kdu.ac.jp (M.M.); 3Department of Health and Nutrition, Faculty of Human Health, Kanazawa Gakuin University, 10 Sue-machi, Kanazawa 9201392, Ishikawa, Japan; t-takahasi@kanazawa-gu.ac.jp; 4Department of Disaster Medicine and Dental Sociology, Graduate School of Dentistry, Kanagawa Dental University, 82 Inaoka, Yokosuka 2388580, Kanagawa, Japan; fuchida@kdu.ac.jp; 5Nakajima Internal Medicine Clinic, 1-17 Yonegahamadori, Yokosuka 2380011, Kanagawa, Japan; toyoda@nakajima-naika.com (M.T.); nakajima@nakajima-naika.com (S.N.)

**Keywords:** periodontal disease, diabetes, diabetic retinopathy, bleeding on probing, probing pocket depth, fasting blood sugar

## Abstract

Both periodontal disease and diabetes are common chronic inflammatory diseases. One of the major problems with type 2 diabetes is that unregulated blood glucose levels damage the vascular endothelium and cause complications. A bidirectional relationship between periodontal disease and diabetic complications has been reported previously. However, whether periodontal disease affects the presence of diabetic complications has not been clarified. Therefore, we examined the effect of the periodontal disease status on diabetic complications in patients with type 2 diabetes. Periodontal doctors examined the periodontal disease status of 104 type 2 diabetic patients who visited a private diabetes medical clinic once a month between 2016 and 2018. The subject’s diabetic status was obtained from their medical records. Bayesian network analysis showed that bleeding on probing directly influenced the presence of diabetic retinopathy in type 2 diabetes patients. In addition, bleeding on probing was higher in the diabetic retinopathy group (*n* = 36) than in the group without diabetic retinopathy (*n* = 68, *p* = 0.006, Welch’s *t*-test). Bleeding on probing represents gingival inflammation, which might affect the presence of diabetic retinopathy in type 2 diabetes patients who regularly visit diabetic clinics.

## 1. Introduction

Both periodontal disease and type 2 diabetes are known to be common chronic inflammatory diseases [1,2]. In addition, many studies have reported a bidirectional relationship between periodontal disease and type 2 diabetes [3,4,5]. Epidemiological studies show that diabetics with poor glycemic control have an increased risk of periodontal disease [6]. On the other hand, it has been reported that severe periodontitis adversely affects glycemic control in type 2 diabetic patients [7]. It has become clear that the treatment of periodontal disease and type 2 diabetes has an effect on the pathophysiology of both diseases. Further, it has been reported that hemoglobin A1c (HbA1c) levels are reduced when type 2 diabetic patients are treated for periodontal disease [8,9,10]. Additionally, it was reported that when blood glucose was controlled in type 2 diabetic patients, the bleeding on probing (BOP) value, which represents the inflammation state of periodontal disease, improved [11]. Both periodontal disease and type 2 diabetes have a high prevalence worldwide [12]; however, the mechanism underlying the bidirectional relationship between them remains unclear.

Type 2 diabetes is a disorder of dysregulated blood glucose homeostasis due to a decrease in insulin action [13]. One of the major problems with type 2 diabetes is that unregulated blood glucose levels damage the vascular endothelium and cause complications in diabetics [14]. Diabetes complications include cardiovascular disease, diabetic nephropathy, neuropathy, leg amputation, and diabetic retinopathy [15]. Diabetic retinopathy is one of the most common diabetic complications and, as it progresses, can cause blindness and poor quality of life [16]. Although diabetic retinopathy is a major cause of blindness in the working population, only 35–55% of patients with diabetes undergo regular ophthalmic evaluations. This is because the disease progresses gradually and the patient barely notices the progression of diabetic retinopathy [16]. Therefore, it is important to prevent the development of diabetic retinopathy in diabetic patients.

It has recently been shown that there is a correlation between the severity of periodontal disease and diabetic retinopathy and that many diabetic retinopathy patients have periodontal disease [17,18]. However, it remains unclear as to whether periodontal disease affects the presence of diabetic retinopathy. Commisso et al. reported that poor oral health care was observed in the diabetic population [19]. It is known that effective oral health behaviors, such as brushing teeth two or more times a day, improve the condition of periodontal disease [20]. However, the relationship between the toothbrushing habits of type 2 diabetic patients and the pathological condition of diabetes has not been fully clarified. Therefore, the purpose of this study was to clarify the effect of periodontal disease status on diabetic complications, including diabetic retinopathy, in type 2 diabetic patients who visit our hospital once a month for diabetes management. Furthermore, this study also aimed to examine the relationship between tooth brushing habits and the pathophysiology of diabetes in patients with type 2 diabetes.

## 2. Methods

### 2.1. Study Population

We recruited 104 (45 men and 59 women) diabetic patients who visited a private diabetes medical clinic in Yokosuka City (Nakajima Internal Medicine Clinic) every month between October 2016 and August 2018. Patients with type 1 diabetes, toothless patients, patients who did not visit the diabetic department monthly, and those who did not consent to the study were excluded. The protocol of the present study was approved by the Ethics Committee of Kanagawa Dental University in 2016 (approval number: 359) and was conducted in accordance with the Helsinki Declaration of 1975, as revised in 2013. The study was registered with the Clinical Trial Registry of the University Hospital Medical Information Network Clinical Trials Registry (ID: UMIN000024627). On implementation, all participants were given written and verbal explanations regarding the purposes and methods of the study, any potential risks, potential benefits, protection of personal information, and freedom of consent and withdrawal. Subsequently, all participants signed the informed consent form.

### 2.2. Medical Examination

Data regarding age, height, weight, duration of diabetes, presence of diabetic retinopathy, and the presence of diabetic nephropathy were collected from the subject’s medical records. For each subject, the diagnosis of diabetic retinopathy was made by an ophthalmologist based on the Davis classification as follows: no diabetic retinopathy (NDR); simple diabetic retinopathy (SDR); pre-proliferative diabetic retinopathy (pre-PDR); and proliferative diabetic retinopathy (PDR) [21]. Peripheral blood samples were collected by a nurse who specializes in diabetes during the routine monthly medical consultations. HbA1c levels were measured by an HbA1c analyzer (ADAMS^TM^ A1c HA-8180, ARKRAY Inc., Kyoto, Japan), and fasting blood sugar (FBS) levels were measured by a glucose analyzer (Glutest mint, SANWA KAGAKU KENKYUSHO Co., LTD., Aichi, Japan). The remaining peripheral blood was used to measure creatinine and triglyceride levels in a clinical laboratory (SRL, Inc., Tokyo, Japan).

### 2.3. Periodontal Examination

The periodontal disease status test was performed at the diabetic medical clinic on the day of the subject’s visit by three trained periodontists. The probing pocket depth (PPD) and BOP were recorded with a manual probe (No.9550, YDM Corporation, Tokyo, Japan) at four points (mesial-buccal, mid-buccal, distal-buccal, and mid-lingual) for all teeth. Measurements of PPD and BOP were performed by three periodontists and then calibrated by a periodontist with the longest clinical experience (M.M.). Tooth mobility was measured for each tooth in the oral cavity. The Miller tooth mobility scale (0 degree, 1 degree, 2 degree, and 3 degree) was used to classify the degree of tooth mobility [22]. The presence of dental plaque on the surface adjacent to each tooth was determined by scratching the adjacent surface on the gingival margin with an explorer. Adherence of dental plaque to the adjacent surface even at one place led to the plaque adhesion on a surface that is adjacent to the tooth to be judged as “Yes”. The number of times patients brushed their teeth per day and whether they underwent regular dental check-ups were confirmed by a questionnaire that was completed by the subjects.

### 2.4. Bayesian Network

A Bayesian network is a directed acyclic graph that is composed of a set of variables {*X*_1_*, X*_2_*,...,X_N_*} and a set of directed edges between the variables [23]. Bayesian networks are very successful in probabilistic knowledge representation and reasoning. In Bayesian networks, the joint probability distribution function of all nodes can be calculated as follows:(1)$$P\;(X_{1},\;X_{2},\;\ldots ,\;X_{N})\;=\;{\prod_{i=1}^{N}P\;(Xi|\;Pai)}_{\;}$$
where *Pai* is the set of random variables whose corresponding nodes are parent nodes of *Xi*.

A Bayesian network contains two elements: structure and parameters. Each arc begins at a parent node and ends at a child node. *Pa* (*X*) represents the parent nodes of node *X*. *X*_1_ is the root node because it has no input arcs. Root nodes have prior probabilities. Each child node has conditional probabilities based on the combination of states of its parent nodes.

Although this study was a cross-sectional study, it was analyzed by a Bayesian network with reference to previous studies by Tsuruta et al. [24].

### 2.5. Statistical Analysis

All statistical analyses were performed using JMP version 12 (SAS Institute Japan, Tokyo, Japan) and R version 3.2.0 (The R Project for Statistical Computing, Vienna, Austria, 2013). Results are expressed as the mean and the standard error of the mean (SEM). Comparisons between the two groups were analyzed using Welch’s *t*-test. Spearman’s rank correlation was employed to analyze the statistical significance of the correlation between two variables. Causal effects between variables were calculated using Bayesian network analysis. *p*-Values less than 0.05 were considered statistically significant.

## 3. Results

### 3.1. Subject Characteristics

The characteristics of the subjects included in this study are shown in Table 1. The mean age of the subjects was 70.0 ± 1.22 years (range: 23–86 years). In terms of dental characteristics, the mean number of teeth was 21.4 ± 0.735, and the number of teeth tended to decrease with age. The average BOP was 27.3%, and 13.2% of subjects had an average ratio of a PPD of 4 mm or greater, with most subjects having periodontal disease. Thirty-eight subjects underwent supportive periodontal therapy at the dental clinic, and 66 did not. In terms of medical characteristics, the average duration of diabetes was 13.6 ± 1.02 years. The mean HbA1c was 7.15%, the mean FBS was 147 mg/dL, and the mean serum creatinine concentration was 1.08 mg/dL. In addition, 36 subjects had diabetic retinopathy, 30 had SDR, 4 had pre-PDR, and 2 had PDR. Twelve subjects had diabetic nephropathy and two subjects had diabetic neuropathy. Most of the subjects in this study had a history of diabetes for more than 10 years, and some had diabetic complications.

### 3.2. Hemoglobin A1c in Type 2 Diabetic Patients with Adjacent Dental Plaque Attached to the Tooth Surface

The subjects were divided into a group with dental plaque attached to the adjacent tooth surface (*n* = 69) and a group without adjacent dental plaque attachment (*n* = 35). HbA1c was higher in the group with dental plaque attachment compared to the group without (*p* < 0.05, Welch’s *t*-test, Table 2). No other differences were found between the two groups (Welch’s *t*-test, Table 2).

### 3.3. Fasting Blood Sugar of Type 2 Diabetic Patients Who Brushed Their Teeth more than Twice a Day

The subjects were divided into two groups: those who brushed their teeth less than once per day (*n* = 39) and those who brushed their teeth two or more times per day (*n* = 65). The FBS in the group who brushed their teeth more than twice per day was lower than that in the group who brushed once a day or less (*p* < 0.0001, Welch’s *t*-test, Table 3). No other differences were found between the two groups (Welch’s *t*-test, Table 3).

### 3.4. Bleeding on Probing in Type 2 Diabetic Patients with Diabetic Retinopathy (1)

The subjects were divided into the diabetic retinopathy group (*n* = 36) and the non-diabetic retinopathy group (*n* = 68). The BOP was higher in the group with diabetic retinopathy than in the group without diabetic retinopathy (*p* = 0.006, Welch’s *t*-test, Table 4). No other differences were found between the two groups (Welch’s *t*-test, Table 4).

### 3.5. Bleeding on Probing in Type 2 Diabetic Patients with Diabetic Retinopathy (2)

Using Spearman’s rank correlation, we found that the FBS positively correlated with the ratio of PPD of 4 mm or greater (r_s_ = 0.29, *p* = 0.003, *n* = 104) and BOP (r_s_ = 0.21, *p* = 0.04, *n* = 104) (Table 2). In contrast, FBS was not correlated with the percentage of teeth with more than 1 degree of movement (r_s_ = −0.041, *p* = 0.7, *n* = 104, Table 5) or the number of teeth (r_s_ = 0.034, *p* = 0.7, *n* = 104, Table 5).

### 3.6. Determination of Causal Effects Using Bayesian Network Analysis

Bayesian network analysis showed that the presence of diabetic retinopathy was directly affected by BOP. In addition, the number of times patients brushed their teeth per day was directly affected by HbA1c and FBS.

## 4. Discussion

### 4.1. Effect of Bleeding on Probing on Diabetic Retinopathy

In this study, the subjects with diabetic retinopathy had a higher BOP compared to those without diabetic retinopathy (Table 4). Furthermore, the Bayesian network analysis showed that the presence of diabetic retinopathy was directly affected by the BOP (Figure 1). The BOP indicates the gingival inflammatory conditions caused by periodontal disease [25]. Accordingly, these results indicate that gingival inflammation in diabetic patients might have affected the development of diabetic retinopathy.

The involvement of reactive oxygen species (ROS) is considered to be a factor in which inflammation of the gingiva affected diabetic retinopathy [26,27,28,29,30,31,32,33]. *Porphyromonas*
*gingivalis* (*P. gingivalis*) and *Fusobacterium*
*nucleatum* (*F*. *nucleatum*) are the major periodontal bacteria that cause gingival inflammation [26,27]. It has been reported that diabetic model rats infected with *P**. gingivalis* have increased maxillofacial oxidative stress and decreased gingival microvascular reactivity [28]. Tothova et al. reported that gingival inflammation caused by periodontitis leads to the production of ROS by neutrophils [29]. Furthermore, ROS production by neutrophils against *F*. *nucleatum* has been shown to be higher than that of *P. gingivalis* [30]. Overproduced ROS in the oral cavity can cause oxidative stress and oxidative damage to cells, proteins, and lipids throughout the body, which can lead to systemic disease [31]. Oxidative stress has been shown to reduce neuroretinal function [32]. In addition, oxidative stress has been found to be associated with the accelerated onset of diabetic retinopathy [33]. Therefore, ROS produced during inflammation of the gingiva in type 2 diabetic patients might induce oxidative stress and contribute to the development of diabetic retinopathy. Suppressing the production of ROS by treating periodontal disease might reduce the development of diabetic retinopathy in diabetic patients.

Interleukin-17 (IL-17), which is secreted by IL-17-producing cells (T helper-17 cell: Th-17), causes inflammation and is involved in the development of chronic diseases such as autoimmune diseases [34]. IL-17 and Th-17 cells are present in human periodontal disease lesions and cause gingival inflammation [35]. Patients with a high BOP ratio have been found to have higher IL-17A concentrations in the saliva and gingival crevicular fluid compared with those without periodontal disease [36]. Furthermore, patients with periodontal disease have been found to have more Th-17 cells that produce IL-17 in the serum compared to healthy individuals [37]; in addition, the serum IL-17 concentration decreased due to a decrease in the BOP ratio in periodontal patients [38]. IL-17A has been reported to exacerbate diabetic retinopathy by impairing the function of Muller cells, which are the major glial cells in the retina [39]. Furthermore, IL-17A receptors have been found to be expressed in Muller glial cells, retinal endothelial cells, and photoreceptors [40]. Increased production of IL-17A in the periodontal lesions of patients with type 2 diabetes might have affected the development of diabetic retinopathy via the IL-17A receptor in the retina. Therefore, suppression of gingival inflammation with proper periodontal treatment might reduce IL-17 production in the gingiva and reduce the development of diabetic retinopathy in diabetic patients.

### 4.2. Relationship between Glycemic Control and Oral Hygiene Behavior

The Bayesian network analysis showed that the number of times patients brushed their teeth per day was directly affected by the HbA1c and FBS (Figure 1). In addition, HbA1c was higher in the group with dental plaque on the adjacent surfaces of the teeth compared to the group without adjacent dental plaque (Table 2). The FBS in subjects who brushed more than twice a day was lower than that of the subjects who brushed once a day or less (Table 3). It is well known that the amount of plaque on the tooth surface increases when the number of times toothbrushing is performed decreases [41]. Furthermore, HbA1c tended to be lower in the group who brushed more than twice a day than in the group who brushed once a day or less (*p* = 0.05, Table 3). From these findings, it is inferred that not only do FBS and HbA1c directly affect the number of times that diabetic patients brush their teeth per day, but also that oral hygiene habits and oral hygiene in diabetic patients are related to glycemic control.

Health literacy is the ability of an individual to obtain, understand, evaluate, and use health information to make decisions regarding the treatment of illnesses and overall health, and to maintain and improve the quality of life [42]. Sense of coherence (SOC) is a personality trait that allows individuals to adapt to and cope with stress to promote their health [43]. Recently, it has been reported that health literacy and SOC influence the pathophysiology of chronic diseases [42,44]. Patients with type 2 diabetes who have poor glycemic control have been shown to have poor health literacy [45]. Since SOC is indirectly involved in the glycemic control of patients with type 2 diabetes, it is necessary to increase SOC to improve glycemic control in such patients [46,47]. In addition, it has been reported that health literacy and SOC also affect oral hygiene behavior. Those who brush their teeth more than twice a day had a higher health literacy [48,49]. It has been reported that subjects who brush their teeth more than twice a day had higher SOC scores [49]. Inferring from these reports, the diabetic patients in this study with a higher health literacy and SOC would have been aware of not only blood glucose control, but also oral health improvement and behavior. Moreover, the ability to maintain and promote the health of diabetics would have been associated with glycemic control and oral hygiene habits. As a result, FBS and HbA1c would have had an effect on the number of times toothbrushing was performed per day, even in the Bayesian network analysis (Figure 1). Patients with type 2 diabetes who brush their teeth less often (per day) and have a large amount of plaque in the oral cavity might have poor glycemic control. Therefore, dentists and dental hygienists need to educate type 2 diabetics who are not interested in oral health in order to improve their health literacy and SOC, which might improve glycemic control in type 2 diabetics.

### 4.3. Relationship Between Periodontal Inflammation and Fasting Blood Sugar

In this study, there was a correlation between FBS and the ratio of both PPD of 4 mm or greater and BOP (Table 5). Similar to BOP, a PPD of 4 mm or greater indicates inflammation of the periodontal tissue [50]. Therefore, this relationship between periodontal inflammation and FBS in patients with type 2 diabetes was expected. There are many reports regarding the relationship between periodontal disease and FBS. Bleeding on probing (BOP) and PPD of 4 mm or greater were associated with fasting blood glucose [51]. For example, Joshipura et al. reported that the FBS of diabetic patients was reduced due to reduced periodontal pockets and improved BOP [52]. Further, Choi et al. reported that the FBS increased as the PPD increased [53]. Periodontal disease increases insulin resistance and raises blood sugar in diabetic patients [54]. Moreover, Colombo et al. reported that periodontal disease model rats had elevated plasma concentrations of tumor necrosis factor-α and increased insulin resistance [55]. The subjects in the present study might have also had high FBS levels due to periodontal inflammation causing insulin resistance. Treatment and education to reduce periodontal inflammation in dental clinics would help to control blood glucose levels in patients with type 2 diabetes.

### 4.4. Considerations

To the best of our knowledge, this study is the first to clarify the effect of periodontal disease on diabetic retinopathy in type 2 diabetic patients, as assessed by the diabetes department. However, this study has some limitations. First, we could not determine the smoking history of subjects because this was not described in the diabetes medical charts of the subjects. Second, we were unable to obtain data on the subjects’ insulin injection status and medication status. As such, future studies will need to consider the association between these data and the effect of periodontal disease on diabetic retinopathy. Third, analysis of the Bayesian network revealed a causal relationship between multiple items. However, since the main purpose of this study was to clarify the effect of periodontal disease on diabetic complications, we considered only the causal relationship between the two diseases.

## 5. Conclusions

In conclusion, the BOP of subjects with diabetic retinopathy was higher than that of the subjects without diabetic retinopathy. Moreover, the gingival inflammation exhibited by BOP affected the presence of diabetic retinopathy in type 2 diabetics who regularly visit diabetic clinics. Controlling gingival inflammation by treating periodontal disease in dental clinics might suppress the development of diabetic retinopathy in patients with type 2 diabetes.

## Figures and Tables

**Figure 1 jcm-09-03234-f001:**
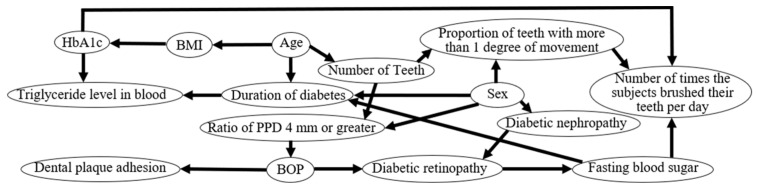
Bayesian network showing the causal effects among the following parameters: gender, age, number of teeth, bleeding on probing (BOP), ratio of probing pocket depth (PPD) of 4 mm or greater, proportion of teeth with more than 1 degree of movement, dental plaque adhesion, number of times the patients brushed their teeth per day, duration of diabetes, body mass index (BMI), fasting blood sugar, hemoglobin A1c (HbA1c), presence of diabetic retinopathy, presence of diabetic nephropathy, and triglyceride level in the blood. Causes and effects are indicated by arrowheads and lines, respectively.

**Table 1 jcm-09-03234-t001:** Characteristics of the subjects in this study.

Total number	*n* = 104
Sex (*n*): Female/male	59/45
Age (year): Mean ± SEM	70.0 ± 1.22
Age (year): Minimum/maximum	23/86
Number of teeth: Mean ± SEM	21.4 ± 0.735
BOP (%)	27.3 ± 1.74
Ratio of PPD, 4 mm or greater (%)	13.2 ± 1.61
Percentage of teeth with more than 1 degree of movement (%)	9.90 ± 1.74
Plaque adhesion on a surface that is adjacent to the tooth (*n*): Yes/no	69/35
Brushing the teeth 2 or more times a day: Yes/no	65/39
SPT in the dental clinic: Yes/no	38/66
Duration of diabetes (years): Mean ± SEM	13.6 ± 1.02
BMI (%): Mean ± SEM	29.4 ± 0.467
FBS (mg/dL): Mean ± SEM	147 ± 4.95
HbA1c (%): Mean ± SEM	7.15 ± 0.0948
Serum creatinine (mg/dL)	1.08 ± 0.178
Diabetic nephropathy: Yes/no	12/92
Diabetic retinopathy: Yes/no	36/68
Diabetic retinopathy: PDR/total number	30/36
Diabetic retinopathy: pre-PDR/total number	4/36
Diabetic retinopathy: PDR/total number	2/36
Diabetic neuropathy: Yes/no	2/102
Serum LDL cholesterol (mg/dL): Mean ± SEM	113 ± 3.09
Serum HDL cholesterol (mg/dL): Mean ± SEM	57.3 ± 1.48
Serum triglyceride (mg/dL): Mean ± SEM	161 ± 1.48

BOP, Bleeding on probing; PPD, probing pocket depth; SPT, supportive periodontal therapy; BMI, body mass index; FBS, fasting blood sugar; HbA1c, hemoglobin A1c; NDR, no diabetic retinopathy; SDR, simple diabetic retinopathy; PDR, proliferative diabetic retinopathy; LDL, low-density lipoprotein; HDL, high-density lipoprotein; SEM, standard error of the mean.

**Table 2 jcm-09-03234-t002:** Characteristics of the groups with and without adjacent dental plaque attachment.

Variable	Adjacent Dental Plaque Attachment (*n* = 69)	No Adjacent Dental Plaque Attachment (*n* = 35)	*p*-Value *
Duration of diabetes (years)	14.3 ± 8.44	12.3 ± 13.4	0.4
BMI (%)	29.5 ± 4.69	29.1 ± 4.97	0.7
FBS (mg/dL)	154 ± 49.1	134 ± 51.2	0.06
HbA1c (%)	7.27 ± 1.01	6.90 ± 0.835	<0.05
Creatinine (mg/dL)	1.25 ± 2.21	0.735 ± 0.190	0.06
LDL cholesterol (mg/dL)	112 ± 32.5	113 ± 29.7	0.9
HDL cholesterol (mg/dL)	55.6 ± 14.1	60.5 ± 11.6	0.1
Triglyceride (mg/dL)	164 ± 88.4	156 ± 112	0.8

Data are presented as the mean ± standard error of the mean. BMI, body mass index; FBS, fasting blood sugar; HbA1c, hemoglobin A1c; Creatinine, serum creatinine; LDL, low-density lipoprotein; HDL, high-density lipoprotein; Triglyceride: serum triglyceride. * Welch’s *t*-test. *p*-values < 0.05 are considered statistically significant.

**Table 3 jcm-09-03234-t003:** Characteristics of the groups who brushed their teeth <1 and ≥2 times per day.

Variable	Group Who Brushed Their Teeth ≥ 2 Times per Day (*n* = 65)	Group Who Brushed Their Teeth < 1 Time per Day (*n* = 39)	*p*-Value *
Duration of diabetes (years)	12.7 ± 10.7	15.1 ± 9.70	0.2
BMI (%)	28.9 ± 4.66	30.3 ± 4.88	0.2
FBS (mg/dL)	128 ± 31.4	179 ± 60.0	<0.0001
HbA1c (%)	7.00 ± 0.906	7.39 ± 1.03	0.05
Creatinine (mg/dL)	0.945 ± 1.56	1.31 ± 2.17	0.4
LDL cholesterol (mg/dL)	115 ± 32.0	108 ± 30.4	0.3
HDL cholesterol (mg/dL)	59.1 ± 13.8	54.2 ± 16.8	0.1
Triglyceride (mg/dL)	153 ± 93.2	173 ± 102	0.3

Data are presented as the mean ± standard error of the mean. BMI, body mass index; FBS, fasting blood sugar; HbA1c, hemoglobin A1c; Creatinine, serum creatinine; LDL, low-density lipoprotein; HDL, high-density lipoprotein; Triglyceride, serum triglyceride. * Welch’s *t*-test. *p*-values < 0.05 are considered statistically significant.

**Table 4 jcm-09-03234-t004:** Characteristics of the groups with and without diabetic retinopathy.

Variable	Diabetic Retinopathy (*n* = 36)	No Diabetic Retinopathy (*n* = 68)	*p*-Value *
Number of teeth	20.7 ± 8.50	21.7 ± 6.94	0.5
BOP (%)	34.1 ± 18.1	23.7 ± 16.6	0.006
Ratio of PPD, 4 mm or greater (%)	17.2 ± 19.0	11.1 ± 14.6	0.09
Proportion of teeth with more than 1 degree of movement (%)	12.0 ± 24.6	8.76 ± 12.7	0.5
Number of times the subjects brushed their teeth per day	1.53 ± 0.774	1.78 ± 0.666	0.1

Data are presented as the mean ± standard error of the mean. BOP, bleeding on probing; PPD, probing pocket depth. * Welch’s *t*-test. *p*-values < 0.05 are considered statistically significant.

**Table 5 jcm-09-03234-t005:** Correlations among fasting blood sugar, the PPD ratio (4 mm or greater), and BOP.

Variable	Fasting Blood Sugar (mg/dL)		
	r_s_ *	*p*-Value	*n*
Ratio of PPD 4 mm or greater (%)	0.29	0.0029	104
BOP (%)	0.21	0.037	104
Proportion of teeth with more than 1 degree of movement (%)	−0.041	0.68	104
Number of teeth	0.034	0.73	104

PPD, Probing pocket depth; BOP, bleeding on probing. * Spearman’s rank correlation coefficient. *p*-values < 0.05 are considered statistically significant.

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
