# Peer review of "Effect of Periodontal Disease on Diabetic Retinopathy in Type 2 Diabetic Patients: A Cross-Sectional Pilot Study"

_jcm, 2020, doi:10.3390/jcm9103234_

Round 1
Reviewer 1 Report
This is an interesting and well written manuscript.
Major concern:
This is a cross-sectional study and therefore, even though the Bayesian network analysis indicates a certain cause and effect relationship with statistical significance, the result and conclusion must be carefully stated particularly in regards to cause and effect. In addition, some arrows in Figure 1 simply do not make sense. For example, how could HbA1c levels or fasting glucose levels influence the number of tooth brushings per day which is indicated in Figure 1? Another example is the effect of BOP on dental plaque adhesion. Please clarify these issues.
Minor concerns:
- Page 3, Line 102: Please revise buccal-mesial to mesial-buccal, buccal-distal to distal-buccal.
- Page 3, Line 108-109: Please add more details on this analysis.
- Throughout the text: Please change PPD of 4mm or more to PPD of 4mm or greater.
- Page 6, Lines 187-189: Please rephrase this paragraph or delete this since it is in the Discussion section.
- Page 6, Line 197-198: Please add a reference here.
- Page 7, Line 223: please change “should” to “might”.
- Page 7, Lines 227-228: Please rephrase this sentence
- Page 7, Line 237-238: Please rephrase this sentence
- Page 7, Lines 252-254: Please rephrase this sentence
Author Response
Response to Reviewer 1’s Comments
“This is a cross-sectional study and therefore, even though the Bayesian network analysis indicates a certain cause and effect relationship with statistical significance, the result and conclusion must be carefully stated particularly in regards to cause and effect. In addition, somearrows in Figure 1 simply do not make sense. For example, how could HbA1c levels or fasting glucose levels influence the number of tooth brushings per day which is indicated in Figure 1? Another example is the effect of BOP on dental plaque adhesion. Please clarify these issues.”
We thank for appropriate advice from you very much. The corrected parts are marked with yellow lines in the manuscript.
According to Reviewer 1’s suggestion, we have revised the manuscript.
This study is a cross-sectional study, but there is a previous study that analyzed the data of the cross-sectional study with a Bayesian network. Therefore, we introduced the previous research and added it to the reference (Line 124-125, Page 3. References No. 24).
The analysis result by the Bayesian network corrected the texts of "Discussion".
The purpose of this study is to clarify the effect of periodontal disease on diabetic complications and the relationship between diabetes and oral hygiene habits. Therefore, we did not consider all the causal relationships revealed by the Bayesian network. Only the causal relationship between the two diseases, diabetes and periodontal disease, was considered. We mentioned that in the “Considerations” (Line 298-300, Page 8).
Minor concerns:
“Page 3, Line 102: Please revise buccal-mesial to mesial-buccal, buccal-distal to distal-buccal.”
According to Reviewer 1’s suggestion, we revised buccal-mesial to mesial-buccal, buccal-distal to distal-buccal (Line 103-104, Page 3).
“Page 3, Line 108-109: Please add more details on this analysis.”
According to Reviewer 1’s suggestion, we have detailed the Bayesian network analysis in our manuscript and added reference (Line 115-123, Page 3).
“Throughout the text: Please change PPD of 4mm or more to PPD of 4mm or greater.”
According to Reviewer 1’s suggestion, in the whole text, PPD of 4mm or more was corrected to PPD of 4mm or greater (yellow lines).
“Page 6, Lines 187-189: Please rephrase this paragraph or delete this since it is in the Discussion section.”
According to Reviewer 1’s suggestion, we have deleted this paragraph.
“Page 6, Line 197-198: Please add a reference here.”According to Reviewer 1’s suggestion, we have added references (Line 215, Page 7).
“Page 7, Line 223: please change “should” to “might”.”
According to Reviewer 1’s suggestion, we changed “should” to “might” (Line 240, Page 7).
“Page 7, Lines 227-228: Please rephrase this sentence.”
According to Reviewer 1’s suggestion, Please rephrase this sentence (Line 244, Page 7).
“Page 7, Line 237-238: Please rephrase this sentence.”
According to Reviewer 1’s suggestion, Please rephrase this sentence (Line 252-255, Page 7).
“Page 7, Lines 252-254: Please rephrase this sentence.”
According to Reviewer 1’s suggestion, Please rephrase this sentence (Line 268-271, Page 8).

Reviewer 2 Report
This manuscript attempts to describe a relationship between the presence of inflammatory periodontal disease, i.e., gingivitis and periodontitis, and diabetic retinopathy. The authors have done a nice job of collecting data and analysis of the data. However, they must be cautioned in that bleeding on probing (BOP) is simply a surrogate measurement of the presence of gingival inflammation and is not, in itself, a factor in diabetic retinopathy. Having said that, the presence of chronic periodontal inflammation, indicated by BOP, may be a risk factor in diabetic retinopathy.
Thus, in light of what this reviewer has noted, I would suggest the authors re-read their manuscript and emphasize the role of inflammation as a risk factor – as indicated by increased BOP in the diabetic retinopathy group.
In the abstract, the authors state that “bleeding on probing may affect the presence of diabetic retinopathy.” However, as stated in line 47, bleeding on probing (BOP) is an indicator of inflammation. The actual BOP affects nothing; it is simply a surrogate measurement for the presence of inflammation. Thus, the statement must be changed, in both the abstract and conclusions section, to reflect inflammation as a possible risk factor for diabetic retinopathy.
In the Materials and Methods section there is no indication that the clinical examiners were calibrated for the periodontal probing depth and bleeding on probing measurements. Further, there is no mention of a tooth mobility scale and yet in Table 1 the authors mention "percentage of teeth with more than 1 degree of movement". What does 1 degree of movement mean - how does this translate into Class I, II, III mobility? Was a mobility scale used, such as the Miller Tooth Mobility Scale (Miller PD Jr. A Classification of Marginal Tissue Recession. Int J Perio Restorative Dent. 1985;5(2):9-13. Wu C-P, et al. Quantitative Analysis of Miller Mobility Index For The Diagnosis of Moderate to Severe Periodontitis - A Cross-Sectional Study. J Dent Sci. 2018;13:43-47.). Lastly, the authors do not state whether the determination of plaque was an all-or-nothing measurement in the M & M section; however, they do state yes/no in Table 1. The yes/no plaque determination should also be stated in the M & M section.
Lines in all Tables should be justified to left margin rather than each line being centered.
Author Response
Response to Reviewer 2’s Comments
“This manuscript attempts to describe a relationship between the presence of inflammatory periodontal disease, i.e., gingivitis and periodontitis, and diabetic retinopathy. The authors have done a nice job of collecting data and analysis of the data. However, they must be cautioned in that bleeding on probing (BOP) is simply a surrogate measurement of the presence of gingival inflammation and is not, in itself, a factor in diabetic retinopathy. Having said that, the presence of chronic periodontal inflammation, indicated by BOP, may be a risk factor in diabetic retinopathy.
Thus, in light of what this reviewer has noted, I would suggest the authors re-read their manuscript and emphasize the role of inflammation as a risk factor – as indicated by increased BOP in the diabetic retinopathy group. In the abstract, the authors state that “bleeding on probing may affect the presence of diabetic retinopathy.” However, as stated in line 47, bleeding on probing (BOP) is an indicator of inflammation. The actual BOP affects nothing; it is simply a surrogate measurement for the presence of inflammation. Thus, the statement must be changed, in both the abstract and conclusions section, to reflect inflammation as a possible risk factor for diabetic retinopathy.”
We thank for appropriate advice from you very much. The corrected parts are marked with yellow lines in the manuscript.
According to Reviewer 2’s suggestion, we have revisited the Abstract and Conclusion that BOP represents gingival inflammation (Line 31-32, Page 1, Line 304, Page 9).
“In the Materials and Methods section there is no indication that the clinical examiners were calibrated for the periodontal probing depth and bleeding on probing measurements.”
According to Reviewer 2’s suggestion, we have described the method of periodontal probing depth and bleeding on probing calibration in the Methods (Line 104-106, Page 3).
“Further, there is no mention of a tooth mobility scale and yet in Table 1 the authors mention "percentage of teeth with more than 1 degree of movement". What does 1 degree of movement mean – how does this translate into Class I, II, III mobility? Was a mobility scale used, such as the Miller Tooth Mobility Scale (Miller PD Jr. A Classification of Marginal Tissue Recession. Int J Perio Restorative Dent.1985;5(2):9-13. Wu C-P, et al. Quantitative Analysis of Miller Mobility Index For The Diagnosis of Moderate to Severe Periodontitis - A Cross-Sectional Study. J Dent Sci. 2018;13:43-47.).”
According to Reviewer 2’s suggestion, we have added a description of tooth mobility to Methods. We have also added references introduced by Reviewer 2 (Line 106-108, Page 3, Line 370-372, Page 10).
“Lastly, the authors do not state whether the determination of plaque was an all-or-nothing measurement in the M & M section; however, they do state yes/no in Table 1. Theyes/no plaque determination should also be stated in the M & M section.”
According to Reviewer 2’s suggestion, we have described in the Methods how to determine if there is plaque adhesion (Line 109-111, Page 3).
“Lines in all Tables should be justified to left margin rather than each line being centered.”
According to Reviewer 2’s suggestion, we have all the table lines aligned to the left margin.
